# Encapsulation of Essential Oils Using Hemp Protein Isolate–Gallic Acid Conjugates: Characterization and Functional Evaluation

**DOI:** 10.3390/polym17131724

**Published:** 2025-06-20

**Authors:** Xinyu Zhang, Haoran Zhu, Feng Xue

**Affiliations:** 1School of Pharmacy, Nanjing University of Chinese Medicine, Nanjing 210023, China; aphrodite0513@163.com (X.Z.); zhuhaoran031014@163.com (H.Z.); 2Jiangsu Key Laboratory of Medicinal Substance and Utilization of Fresh Chinese Medicine, Nanjing University of Chinese Medicine, Nanjing 210023, China

**Keywords:** hemp protein, conjugates, essential oil, microcapsule, physicochemical properties

## Abstract

Essential oils (EOs) represent natural bioactive agents with broad applications; however, their industrial utilization is often hampered by inherent volatility and instability, which current encapsulation methods struggle to overcome due to limitations such as reliance on synthetic surfactants. Proteins, owing to their amphiphilic nature, serve as materials for EOs microencapsulation, particularly when chemically modified. Building upon our previous work demonstrating improved emulsifying properties of hemp seed protein isolate (HPI) through covalent modification with gallic acid (GA), this study investigated its efficacy for essential oil encapsulation. This study developed a novel microencapsulation system utilizing conjugates of HPI and GA for stabilizing six essential oils (lemon, grapefruit, camellia, fragrans, oregano, and mustard). The microcapsules exhibited encapsulation efficiencies (EE) ranging from 40% to 88%, with oregano oil demonstrating superior performance due to carvacrol’s amphiphilic surfactant properties. Advanced characterization techniques revealed that high-EE microcapsules displayed compact morphologies, enhanced thermal stability, and reduced surface oil localization. Release kinetics followed either the Peppas or Weibull model, with oregano microcapsules achieving sustained release via matrix erosion mechanisms. Antioxidant assays and antimicrobial tests demonstrated multifunctional efficacy, where oregano microcapsules exhibited the highest radical scavenging and antimicrobial activity. These findings establish HPI-GA conjugates as unique dual-functional emulsifier-encapsulants, offering a sustainable and effective platform to enhance EO stability and bioactivity, particularly for applications in food preservation and pharmaceutical formulations.

## 1. Introduction

Essential oils (EOs), volatile aromatic compounds extracted from various plant organs, are valued across the food, pharmaceutical, and agricultural industries for their potent antimicrobial and antioxidant properties [1,2]. Their bioactive constituents, primarily phenolics and terpenoids, enable their applications as natural preservatives, antibiotic alternatives, and biopesticides [3,4]. However, the widespread industrial utilization of EOs is significantly hindered by their intrinsic volatility and susceptibility to degradation under environmental factors like light, heat, and oxygen.

To overcome these limitations, encapsulation strategies are essential. Techniques such as microencapsulation [5], emulsification [6], and incorporation into edible films [7] or porous substrates (e.g., metal–organic frameworks) [8] have been developed to protect EOs, enhance stability, prolong functionality, and enable controlled release. The choice of encapsulation material critically determines system performance. Biomaterials used range from polysaccharides and lipids to proteins and inorganic matrices [9]. Among these, proteins are particularly promising due to their amphiphilic nature and the abundance of functional groups (e.g., amino, carboxyl, hydroxyl), facilitating strong interactions with both hydrophobic and hydrophilic EO components through hydrophobic associations, hydrogen bonding, and electrostatic forces [10]. Current protein-based systems often utilize configurations like single proteins, protein–polysaccharide complexes (formed via Maillard reactions or electrostatic interactions), or assemblies of multiple proteins with adjustable molecular linkages (crosslinking density) [10].

Despite progress, many encapsulation systems face limitations, including reliance on synthetic polymers, complex multi-step fabrication, suboptimal encapsulation efficiency, or insufficient protection against degradation. This creates a compelling need for sustainable, high-performance natural materials. Hempseed protein isolate (HPI), derived as a byproduct from industrial hemp oil processing, presents an attractive, underexploited resource offering significant potential for sustainability and waste valorization [11]. While HPI possesses useful functional properties like film-forming and gelling capacity [12], its effectiveness as an emulsifier for demanding applications like EO encapsulation is limited by inherent structural rigidity and poor water dispersibility, leading to suboptimal interfacial stabilization [13].

A promising strategy to overcome these limitations is covalent conjugation with phenolic compounds. Our recent work pioneered a novel, efficient ultrasonication-induced mechanochemical method to create covalent conjugates between HPI and polyphenols like gallic acid (GA) [14,15]. This approach leverages intense sound waves to generate free radicals, driving rapid covalent bond formation under mild conditions, offering distinct advantages over conventional chemical or enzymatic conjugation routes in terms of speed, efficiency, and potential for scale-up. These HPI–polyphenol conjugates exhibit markedly improved emulsification properties due to enhanced molecular flexibility and interfacial adsorption kinetics. However, a critical knowledge gap exists: the potential of these HPI–GA conjugates to function not only as superior emulsifiers but also as effective dual-functional matrices for the microencapsulation and stabilization of EOs remains unexplored.

Therefore, this study aims to bridge this gap by developing and characterizing a novel microencapsulation platform utilizing HPI–GA conjugates for stabilizing diverse essential oils (lemon, grapefruit, camellia, fragrans, oregano, and mustard). The selected ingredients are all food-approved materials, guaranteeing their biological safety while simultaneously offering well-documented antioxidant and antibacterial properties, according to the existing literature [16,17,18]. We systematically evaluated the physicochemical properties (encapsulation efficiency, morphology, and thermal stability), release kinetics, antioxidant capacity, and antimicrobial efficacy of the resulting microcapsules against foodborne pathogens, specifically targeting enhanced stability and functionality for applications in food preservation and pharmaceutical formulations.

## 2. Materials and Methods

### 2.1. Materials

Hemp seeds (*Cannabis sativa* L.) were obtained from Yunnan Industrial Hemp Co., Ltd. (Kunming, China). The seeds were ground using an electric grinder (Breville BCG300, Botany, Australia) and sieved through a 250 µm mesh to obtain fine powder for subsequent protein extraction. Six commercial essential oils (lemon [*Citrus limon*] (S27236) [19], grapefruit [*Citrus paradisi*] (S27108) [20], camellia [*Camellia oleifera*] (S33125) [21], fragrans [*Osmanthus fragrans*] (S27234) [22], oregano [*Origanum vulgare*] (S27449) [20], and mustard [*Brassica juncea*]) (S27503) [20], DPPH (98%) and ABTS (98%) free radicals were purchased from Shanghai Yuanye Bio-Technology Co., Ltd. (Shanghai, China). Analytical-grade reagents including gallic acid (98%), potassium bromide (KBr, 99%), and absolute ethanol were procured from Sinopharm Chemical Reagent Co., Ltd. (Shanghai, China). All chemicals were used without further purification.

### 2.2. Extraction of Hemp Protein Isolate

The hemp protein isolate (HPI) was prepared according to our previously established protocol [23]. Briefly, hemp seed powder was first defatted using n-hexane (1:10 *w*/*v*) with continuous stirring for 4 h at room temperature. The defatted meal was then subjected to alkaline extraction using Milli-Q water (1:15 *w*/*v*) adjusted to pH 10.0 with 1 M NaOH, followed by continuous stirring for 1 h at 40 °C. After centrifugation at 6000× *g* for 30 min, the supernatant was collected and acidified to pH 4.3 (isoelectric point of hemp protein) using 1 M HCl to precipitate the proteins. The protein precipitate was recovered by centrifugation (6000× *g*, 30 min), washed twice with distilled water, neutralized to pH 7.0, and finally freeze-dried (Labconco FreeZone, −50 °C, 15 Pa, 48 h) to obtain the HPI powder.

### 2.3. Preparation of HPI–Gallic Acid Conjugates

The HPI–gallic acid conjugates were synthesized following our previously established protocol [14]. Briefly, 1 g of HPI and 15 mg gallic acid were dispersed in 100 mL distilled water and homogenized (FM200 homogenizer, FLUKO, Shantou, China) at 12,000 rpm for 1 min. The pH of the resulting dispersion was adjusted to 9.0 using 1 M NaOH, followed by ultrasonication (NingBo Scientz Biotechnology Co., Ningbo, China) with a 0.636 cm diameter titanium probe at 400 W for 20 min under pulsed mode (2 s on/2 s off). To prevent thermal degradation, the sonication process was conducted in an ice-water bath to maintain temperature control. Subsequently, the pH was readjusted to 7.0 using 1 M HCl. The solution was then dialyzed (8–14 kDa) against distilled water at 4 °C for 48 h. Finally, the conjugates were lyophilized and stored at 4 °C for further analysis.

### 2.4. Encapsulation of Essential Oil by Using HPI–Gallic Acid Conjugates

The encapsulation of essential oils was achieved through a two-step homogenization process [24]. Initially, 1 g of HPI–gallic acid conjugates was dispersed in 100 mL of Milli-Q water, followed by the addition of 1 g essential oil. Primary emulsification was performed using a high-speed homogenizer (FM200 homogenizer, FLUKO, Shantou, China) at 12,000 rpm for 5 min to obtain a coarse emulsion. Subsequently, the emulsion was further refined using a high-pressure homogenizer (Ningbo Xinzhi Biotechnology Co., Ltd., Ningbo, China) operating at 50 MPa for three consecutive cycles to achieve uniform droplet size distribution. The resulting emulsion was lyophilized for 48 h, and the freeze-dried product was gently ground to obtain a free-flowing powder. The microcapsule formulations containing lemon, grapefruit, camellia, fragrans, oregano, and mustard essential oils were systematically designated as HG-LEO, HG-GEO, HG-CEO, HG-FEO, HG-OEO, and HG-MEO, respectively, following a nomenclature convention where “HG” represents the wall material (HPI-gallic acid conjugates) and the subsequent three-letter code identifies the encapsulated essential oil.

### 2.5. Determination of Encapsulation Efficiency

The encapsulation efficiency (EE) of the freeze-dried microcapsules was quantified according to a previous method with slight modifications [25]. EE was calculated based on the surface essential oil content and total essential oil content using the following equation: EE (%) = [(Total oil − Surface oil)/Total oil] × 100. For surface oil analysis, 2 g of microcapsules was dispersed in 40 mL n-hexane and stirred continuously (150 rpm) for 1 min. The mixture was then filtered and centrifuged (4000× *g*, 4 °C, 10 min) to separate the oil phase. The supernatant was collected and the solvent was evaporated at 40 °C under reduced pressure. Total oil content was determined by first completely dissolving 2 g of microcapsules in 12 mL of 4 N HCl, followed by extraction with 50 mL n-hexane. After centrifugation under identical conditions (4000× *g*, 4 °C, 10 min), the organic phase was recovered and the solvent was removed as described above. The extracted oils from both procedures were weighed to calculate the encapsulation efficiency. All measurements were performed in triplicate.

### 2.6. Particle Size of Microcapsules in Dispersion

The particle size distribution of the microcapsules was determined by laser diffraction analysis. Briefly, 1 g of microcapsules was uniformly dispersed in 100 mL of distilled water under gentle magnetic stirring (500 rpm) for 5 min to ensure complete dispersion. Particle size measurements were performed using a laser diffraction particle size analyzer (LS 13320, Beckman Coulter Inc., Duarte, CA, USA) with distilled water as the dispersant medium. The instrument parameters were set with refractive indices of 1.33 for the dispersant (water) and 1.50 for the microcapsule particles. Measurements were conducted in triplicate at 25 °C, with the results expressed as volume mean diameter (D [4,3]). All measurements were performed in triplicate.

### 2.7. Fourier Transform Infrared (FTIR) Spectroscopy Analysis of Microcapsules

The dried sample was thoroughly mixed with potassium bromide (KBr) at an approximate ratio of 1:100 (sample:KBr) and finely ground using an agate mortar. The homogeneous mixture was then compressed into a transparent pellet (13 mm diameter) under a hydraulic pressure of 10 MPa for 2 min using a hydraulic press (Specac, Kent, UK). Spectra were acquired using a Fourier transform infrared spectrometer (Nicolet iS5, Thermo Scientific, Waltham, MA, USA) equipped with a deuterated triglycine sulfate detector. Measurements were performed in the mid-infrared region (4000–400 cm^−1^) with a spectral resolution of 4 cm^−1^. Each spectrum represented an average of 32 consecutive scans to improve the signal-to-noise ratio. Background correction was performed using a pure KBr pellet as reference. All measurements were conducted at room temperature (25 ± 1 °C) under controlled humidity conditions (40 ± 5% RH).

### 2.8. Thermal Properties of Microcapsules

Thermogravimetric analysis (TGA) was performed using a Q500 analyzer (TA Instruments, New Castle, DE, USA) to evaluate the thermal decomposition behavior. Approximately 5 mg of the sample was precisely weighed into a ceramic crucible and heated from 30 to 500 °C at a constant heating rate of 10 °C min^−1^ under a nitrogen atmosphere (flow rate: 50 mL min^−1^). The mass loss rate was recorded as a function of temperature. Differential scanning calorimetry (DSC) measurements were conducted using a DSC Q2000 instrument (TA Instruments, USA) to characterize thermal transitions. Samples (5 mg) were hermetically sealed in aluminum pans and subjected to heating from 30 to 250 °C at 10 °C min^−1^ under a nitrogen purge (50 mL min^−1^). Both endothermic and exothermic events were recorded during the heating process. All measurements were performed in triplicate.

### 2.9. Acquiring Microstructure of Microcapsules

The surface morphology of the microcapsules was examined using scanning electron microscopy (SEM; Regulus 8100, Hitachi, Tokyo, Japan). Prior to imaging, samples were mounted on aluminum stubs using conductive carbon tape and sputter-coated with a 10 nm gold–palladium layer (E-1045, Hitachi, Japan) to enhance surface conductivity. Microstructural analysis was performed at an accelerating voltage of 5 kV under high-vacuum conditions (<10^−3^ Pa). Representative images were captured at 5000× magnification to elucidate surface topography and morphological features.

### 2.10. Controlled Release of Essential Oil from Microcapsules

The release rate of encapsulated essential oils was evaluated using an established dialysis method with modifications [16]. Briefly, samples (200 mg) were loaded into pre-hydrated dialysis membranes (molecular weight cutoff: 8–14 kDa) and immersed in 50 mL of ethanolic release medium (50% *v*/*v* ethanol/water) maintained at 37 °C under constant agitation (100 rpm). A standard calibration curve was constructed by preparing serial dilutions of pure essential oil in the release medium, followed by absorbance measurements at 274 nm using a Spark 10M microplate reader (Tecan, Männedorf, Switzerland) with a 96-well plate configuration. All measurements were performed in triplicate.

### 2.11. Measurement of Antioxidant and Antimicrobial Properties

The antioxidant capacity was evaluated using DPPH and ABTS radical scavenging assays following established protocols with modifications [24]. For the DPPH assay, samples were reacted with 0.10 mM DPPH solution in ethanol (final concentration 2.0 mg/mL), while the ABTS assay employed 7 mM ABTS radical cation solution (final concentration 2.0 mg/mL). After 30 min of dark incubation at 25 °C, absorbance was measured at 517 nm (DPPH) or 734 nm (ABTS) using a microplate reader (200 μL aliquots in 96-well plates). All measurements were performed in triplicate.

Antibacterial efficacy was determined against Gram-negative (*Escherichia coli* ATCC 25922) and Gram-positive (*Staphylococcus aureus* ATCC 29213) strains using a modified agar diffusion method [17]. The selected strains serve as both representative models of Gram-negative and Gram-positive bacteria and clinically relevant pathogens [26,27]. Consequently, examining the antimicrobial activity of essential oil microcapsules against these microorganisms provides crucial insights into their efficacy for food preservation and safety applications. Briefly, 25 mL of molten Luria–Bertani agar was inoculated with a 100 μL bacterial suspension (∼10^6^ CFU/mL) before solidification in Petri dishes. Sterile filter paper disks (10 mm in diameter) impregnated with 100 μL sample solution (20 mg/mL in sterile 1% aqueous solution) were aseptically placed on seeded agar. After 24 h of incubation at 37 °C, inhibition zone diameters (including disk diameter) were measured using digital calipers. All measurements were performed in triplicate.

### 2.12. Statistical Analysis

All investigative protocols involving specimen processing and instrumental analyses were systematically executed with three independent replicates (*n* = 3) unless specified differently in particular cases. Numerical results were statistically expressed as mean ± SD (standard deviation) following triplicate verification. For comparative statistical evaluation, we implemented the IBM SPSS Statistics platform (v25.0, Armonk, NY, USA) utilizing a completely randomized experimental design. Intergroup comparisons were conducted through one-way ANOVA complemented with Tukey’s honestly significant difference (HSD) multiple comparison protocol. Probability values reaching the *p* < 0.05 threshold (corresponding to α = 0.05 significance level) at a 95% confidence interval were established as the criterion for determining statistical significance between experimental groups. 

## 3. Results and Discussion

### 3.1. Effect of Essential Oil Type on the Encapsulation Efficiency

The encapsulation efficiency of essential oils by hemp seed protein–gallic acid conjugate as a wall material was observed to range from 40% to 88% (as shown in Figure 1), which was consistent with previously reported values (32–96%) for natural polymers used for essential oil encapsulation [28,29,30,31,32,33]. Thus, the essential oil content per 100 g of HG-OEO, HG-CEO, HG-GEO, HG-LEO, HG-MEO and HG-FEO microcapsules is approximately 47 g, 41 g, 38 g, 36 g, 33 g and 29 g, respectively. The embedding efficiency was found to be influenced not only by the selected wall materials but also significantly associated with the specific essential oil types. Similar phenomena were reported in an earlier study, where the encapsulation efficiency of essential oils by hemp seed protein/ gum Arabic complex was demonstrated to be substantially affected by oil varieties [17]. This variation was primarily attributed to the chemical composition of essential oils. When hydrophilic small-molecule compounds with surface-active properties were present in essential oils, interactions between these components and wall materials were shown to enhance encapsulation performance [34]. Furthermore, the existence of long-chain fatty acids in essential oils was demonstrated to improve emulsion stability through molecular interactions with wall materials, consequently increasing encapsulation efficiency [35]. In the current investigation, oregano essential oil exhibited superior encapsulation efficiency compared to other tested varieties. This phenomenon was hypothesized to be related to its high carvacrol content (>57%) [33,36,37]. The amphiphilic structure of carvacrol, characterized by a benzene ring and hydroxyl group, was suggested to function as a natural surfactant [17,38]. This unique configuration was postulated to facilitate interfacial stabilization between oil and aqueous phases, thereby improving both emulsion stability and encapsulation efficiency.

### 3.2. Effect of Essential Oils on the Microstructure of Capsules in an Aqueous Medium

As shown in Figure 2, the essential oil type significantly influenced the dispersion behavior of microcapsules in the aqueous solution. Microcapsules with higher encapsulation efficiency (e.g., those containing oregano essential oil) exhibited uniform dispersion in water. In contrast, samples with lower encapsulation efficiency (e.g., microcapsules incorporating fragrans and mustard essential oils) primarily existed as large aggregates in the aqueous phase. This phenomenon can be attributed to the formation of a well-adsorbed wall material layer around the essential oil core in high-encapsulation-efficiency samples, which facilitated their aqueous dispersion [24]. Conversely, in low-encapsulation-efficiency samples, the presence of excessive free essential oil on the microcapsule surface promoted interparticle aggregation through wall material interactions. These observations align with those of a previous study investigating the impact of essential oil types on microcapsule microstructure in aqueous systems, where superior encapsulation efficiency was consistently correlated with improved dispersion characteristics [17].

### 3.3. Effect of Essential Oils on the Particle Size of Capsules in Aqueous Phase

As shown in Figure 3, the essential oil-loaded microcapsules exhibited particle sizes ranging from 1 to 25 μm. This size distribution is comparable to other protein-based microencapsulation systems reported in the literature: quinoa protein microcapsules (1–5 μm) [39], rice protein microcapsules (3–10 μm) [40], and plum seed protein microcapsules (0.1–100 μm) [24]. The particle size was significantly influenced by the wall material composition. Notably, polysaccharide-based wall materials generally yielded smaller microcapsules, as evidenced by chitosan microcapsules (200–400 nm) [41] and β-cyclodextrin microcapsules (800 nm) [42]. Our results demonstrated consistency between particle size measurements and microscopic observations. Oregano oil-containing microcapsules showed smaller diameters compared to other essential oil-containing counterparts, indicating that essential oil type significantly affects microcapsule dispersion in aqueous systems. This phenomenon may be attributed to specific molecular interactions between essential oil components and wall materials. A previous study has reported that carvacrol, the major component of oregano oil, can establish strong intermolecular interactions with wall materials, thereby facilitating the formation of smaller microcapsules [43].

### 3.4. Interaction of Essential Oils with the Wall Materials of Capsules

The FTIR spectra revealed significant variations in characteristic absorption bands corresponding to different molecular interactions (Figure 4). In the 1000–1200 cm^−1^ region, the absorption peak attributed to symmetric C-O-C stretching vibrations of fatty acid esters showed marked differences among microcapsules containing different essential oils [44]. Microcapsules with higher encapsulation efficiency, particularly those containing oregano oil, exhibited negligible absorption in this region, indicating complete encapsulation of the essential oil within the wall matrix. Conversely, microcapsules with lower encapsulation efficiency displayed intense absorption peaks, suggesting surface localization of essential oil molecules. These observations align with a previous report demonstrating attenuated absorption in this region when essential oils are effectively encapsulated [24]. The absorption band at 1600 cm^−1^, typically associated with hydrophobic and electrostatic interactions [43,45], showed a positive correlation with encapsulation efficiency in camellia, grapefruit, and lemon essential oil microcapsules. However, the oregano oil microcapsules did not show the strongest absorption, nor did the fragrans essential oil microcapsules display the weakest signal. This anomaly may stem from additional interfacial interactions between surface-localized essential oil molecules and the polymer matrix. A previous study has demonstrated that surface-adsorbed bioactive compounds can significantly alter the characteristic absorption patterns of microcapsules in FTIR spectra [46]. The 2900–3000 cm^−1^ region (C-H stretching vibrations in fatty acids) [47] showed significantly stronger absorption in fragrans essential oil microcapsules, reinforcing the surface localization hypothesis. The 3300–3500 cm^−1^ region, corresponding to O-H stretching vibrations, provided insights into hydrogen bonding interactions. Microcapsules with higher EE consistently exhibited stronger absorption in this region, while those with lower EE showed diminished signals. This correlation establishes hydrogen bonding between wall polymers as a critical determinant of encapsulation performance.

### 3.5. Effect of Essential Oils on the Thermal Properties of Capsules

As shown in Figure 5, the DTG curves exhibited two distinct mass loss stages: the first stage (100–250 °C) corresponded to the volatilization of essential oils [48,49,50], while the second stage (250–450 °C) was attributed to the thermal degradation of the polymer matrix [51,52]. The type of essential oil significantly influenced the thermal stability of the microcapsules. Microcapsules with high encapsulation efficiency showed no prominent mass loss peak in the first stage, indicating that the essential oils were effectively encapsulated within the polymeric core, preventing abrupt release. In contrast, microcapsules with lower encapsulation efficiency displayed a pronounced mass loss peak between 100 and 250 °C, confirming the surface localization of essential oils. Previous research on essential oil microencapsulation have consistently demonstrated that higher encapsulation efficiency leads to a significant increase in the volatilization temperature of encapsulated oils [24]. Furthermore, during the polymer degradation stage, high-encapsulation-efficiency microcapsules exhibited a higher degradation temperature compared to their low-encapsulation-efficiency counterparts, suggesting enhanced thermal stability due to improved encapsulation structure.

As illustrated in Figure 6, the DSC thermograms revealed a distinct endothermic peak between 70 and 100 °C, corresponding to protein denaturation within the microcapsule wall matrix [24]. The denaturation temperature (Td) exhibited significant variation depending on the essential oil type, with high-encapsulation-efficiency microcapsules demonstrating an increase in Td compared to their low-encapsulation-efficiency counterparts. This thermal stabilization effect was particularly pronounced in oregano oil microcapsules, which showed the highest Td (86 °C) among all tested formulations. The elevated thermal resistance can be attributed to enhanced intermolecular interactions between protein chains, as evidenced by FTIR analysis. These findings collectively demonstrate that improved essential oil encapsulation directly correlates with the enhanced thermal stability of the microcapsule system. Previous research on essential oil microencapsulation has consistently demonstrated that higher encapsulation efficiency significantly improves the thermal stability of microcapsules [17].

### 3.6. Effect of Essential Oils on the Microstructure of Capsules

As depicted in Figure 7, the essential oil-loaded microcapsules exhibited a distinctive sponge-like three-dimensional architecture, which aligns with previous findings on thyme oil/whey protein [53] and β-pinene/milk protein/carboxymethyl microcapsules [54]. SEM analysis revealed that the microstructural characteristics were significantly influenced by essential oil type. Microcapsules with lower encapsulation efficiency displayed highly porous structures, resulting from insufficient wall material confinement during the spray-drying process. This porous morphology facilitated the exposure of functional groups (e.g., hydrophobic groups) in wall polymers [17], thereby promoting intermolecular interactions and aggregate formation—a finding consistent with particle size measurements. In contrast, oregano oil microcapsules demonstrated remarkably compact structures with minimal surface porosity, attributable to the following: (1) stronger molecular affinity between oregano oil components (e.g., carvacrol) and wall proteins, and (2) more effective polymer matrix formation during encapsulation. These structural differences explain the superior encapsulation efficiency observed in oregano oil microcapsules compared to other formulations.

### 3.7. Effect of Essential Oils on Their Controlled Release

As shown in Figure 8, the essential oil exhibited a sustained release profile from the microcapsules, with its release rate demonstrating a time-dependent pattern. Our result revealed that the essential oil reached equilibrium release at approximately 260 min, with no significant increase in the release rate observed beyond this time point. These findings align with previous reports on protein-based microcapsules, where hemp protein microcapsules required 210 min to reach equilibrium [17] and soy protein-based systems needed 200 min [55], while plum seed protein microcapsules achieved equilibrium faster at 120 min [24]. This comparative analysis suggests that hemp protein forms more compact microcapsule structures with superior sustained-release properties. Importantly, release kinetics are influenced not only by protein type but also by the structural matrix formed—for instance, protein-based films exhibit much slower release profiles (190 h for soy protein films [16] and 120 h for plum seed protein films [56]). Notably, all microcapsule systems in our study showed less than 80% total oil release, indicating incomplete liberation, likely due to the specific release mechanism in our system where oil diffusion depends primarily on protein dissolution and matrix erosion. This represents a limitation of the current study, and future research should investigate alternative release systems, particularly for food preservation applications where microbial protease activity could potentially enable intelligent microbe-responsive release mechanisms. Such investigations would provide valuable insights for developing smart delivery systems in food applications. As expected, the sustained release rate varied significantly depending on the type of essential oil, with oregano essential oil showing the lowest release rate among all tested samples. This phenomenon may be attributed to its relatively denser microstructure, which could potentially hinder the diffusion process through the microcapsule matrix. Previous studies have similarly demonstrated that the formation of dense polymeric microcapsule structures can effectively retard the release of essential oils [16,56]. Furthermore, the release kinetics were found to be significantly influenced by the type of encapsulated essential oil. Previous research has employed both the Peppas and Weibull models to characterize the release kinetics of essential oils from capsules [56]. As presented in Table 1, the release profiles of lemon and fragrans essential oils from microcapsules followed the Peppas model, as evidenced by their high correlation coefficients (R^2^ > 0.95). The release exponent (n) values exceeding 0.5 for lemon essential oil microcapsules suggest a predominantly Fickian diffusion release mechanism [57]. In contrast, the release patterns of grapefruit, camellia, oregano, and mustard essential oil microcapsules were better described by the Weibull model. Notably, the parameter (b) values greater than 1 for camellia and oregano essential oil microcapsules indicate a combined release mechanism involving both diffusion and polymer swelling [58]. Particularly for oregano essential oil, its release was found to be strongly dependent on wall material erosion due to the exceptionally dense microstructure of the microcapsules, a phenomenon consistent with previous reports on oregano oil encapsulation systems [17].

### 3.8. Effect of Essential Oils on the Antioxidant Properties of Microcapsules

As illustrated in Figure 9, all essential oil microcapsule samples demonstrated free radical scavenging capacity, which can be attributed to two primary factors: (1) the presence of gallic acid, an antioxidant compound [59], in the microcapsule wall material, and (2) the phenolic and terpenoid compounds contained within the essential oils themselves [60,61]. Research investigations have revealed that microencapsulated forms of various essential oils—specifically lemon [62], grapefruit [63], camellia [64], oregano [24], and mustard [65] oils—exhibit notable antioxidant activities. However, the antioxidant performance exhibited complex variations across different encapsulation systems and essential oil types without showing a distinct pattern. This phenomenon likely results from multiple influencing factors, including but not limited to the following: the specific composition of essential oils, their volatility characteristics, the types of free radicals involved, and potential interactions between essential oil components and the wall material matrix [66]. Notably, among all tested samples, oregano essential oil microcapsules displayed superior antioxidant activity. This enhanced performance may be explained by their higher encapsulation efficiency and improved dispersion properties, which collectively increase the probability of contact between active compounds and free radicals. These findings align with those of previous studies demonstrating that increasing the exposure opportunities between antioxidant substances and free radicals can significantly enhance antioxidant efficacy [67,68].

### 3.9. Effect of Essential Oils on the Antimicrobial Properties of Capsules

The inherent hydrophobicity of essential oils enables them to exert antimicrobial effects by altering the permeability of microbial cell membranes [69]. As demonstrated in Figure 10, similarly to the antioxidant properties, the antimicrobial performance of essential oil microcapsules did not follow a distinct pattern, further confirming the multifactorial nature of factors influencing microcapsule antibacterial efficacy. Substantial evidence from the existing literature confirms the antimicrobial efficacy of encapsulated essential oils derived from lemon [70], grapefruit [71], camellia [64], fragrans [72], oregano [73], and mustard [74] sources. Notably, oregano essential oil microcapsules exhibited superior antimicrobial activity, which can be primarily attributed to two key aspects: (1) the intrinsically strong antimicrobial properties of oregano oil itself—previous comparative research has consistently shown that oregano oil possesses significantly greater antimicrobial efficacy than lemon, tsaoko, and grapefruit essential oils [16]; (2) the optimized sustained-release characteristics of oregano oil microcapsules, which enable prolonged and controlled release of active compounds, thereby maintaining effective antimicrobial concentrations over extended periods. These findings corroborate the existing literature, demonstrating that microcapsules with both excellent sustained-release properties and good dispersion characteristics typically achieve enhanced antimicrobial effects [33].

## 4. Conclusions

This study demonstrates that HPI–GA conjugates represent a novel and effective dual-functional platform for the microencapsulation and stabilization of essential oils. Critically, our work establishes HPI–GA conjugates as unique wall materials that simultaneously provide superior emulsification capabilities and robust encapsulation functionality—a combination not widely reported for plant protein–polyphenol systems, particularly those derived from industrial hemp byproducts. This inherent dual action addresses key limitations in current encapsulation matrices, such as reliance on synthetic surfactants or multi-component systems, by offering a sustainable, single-material solution derived from valorized waste.

Our findings confirm the core hypothesis that covalent HPI–GA conjugation creates a matrix capable of effectively entrapping diverse EOs while enhancing their stability and bioactivity. The research reveals that the physicochemical properties of the encapsulated EO itself are crucial determinants of microcapsule performance, influencing encapsulation efficiency, morphology, release kinetics, and functional efficacy. Notably, microcapsules based on HPI–GA conjugates achieved high encapsulation efficiencies and demonstrated controlled release profiles governed by matrix erosion, alongside significantly enhanced antioxidant and antimicrobial activities. These functional outcomes directly fulfill the study’s aim to develop an encapsulation system enhancing EO stability and bioactivity, specifically positioning HPI–GA microcapsules as highly promising for applications demanding sustained antimicrobial and antioxidant effects, such as active food packaging and natural preservative systems.

We acknowledge that this study, conducted under controlled in vitro conditions, has limitations. The performance was not directly compared side-by-side with unencapsulated oils in stability or activity assays, and the complex interactions within real food matrices remain to be evaluated. Future research should prioritize the following: (1) validating efficacy in complex food systems (e.g., evaluating preservation effects in specific matrices like baked goods, meats, or dairy products under varying storage conditions—temperature, humidity, and pH); (2) assessing organoleptic impact to ensure consumer acceptability; and (3) addressing key scale-up challenges, including optimizing conjugation and encapsulation processes for cost-effective manufacturing, ensuring batch-to-batch consistency, and evaluating long-term storage stability under industrial conditions. Overcoming these hurdles is essential for translating this promising technology into commercially viable solutions for the food and pharmaceutical industries.

## Figures and Tables

**Figure 1 polymers-17-01724-f001:**
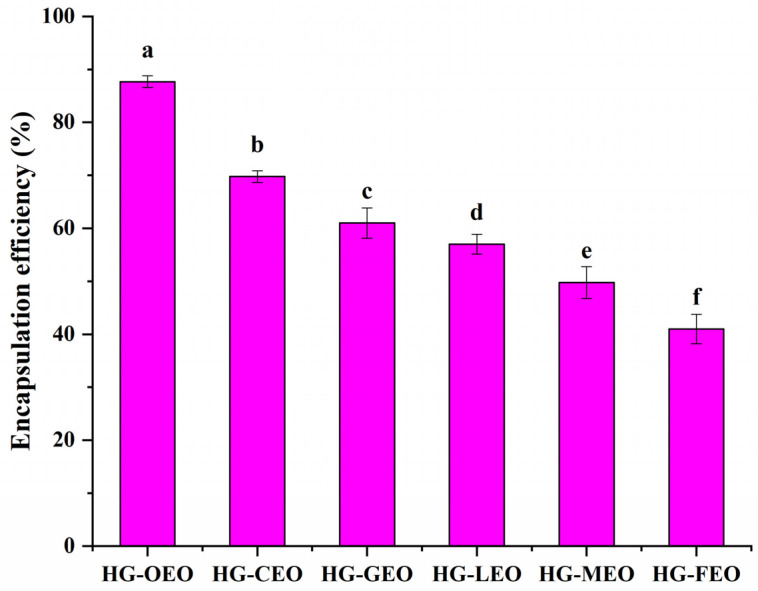
Encapsulation efficiency of essential oil encapsulated in hemp protein isolate (HPI)–gallic acid conjugates. HG: conjugates of hemp protein isolate and gallic acid; LEO: essential oil from lemon; GEO: essential oil from grapefruit; CEO: essential oil from camellia; FEO: essential oil from fragrans; OEO: essential oil from oregano; MEO: essential oil from mustard. Different letters mean that the values showed significant differences (*p* < 0.05).

**Figure 2 polymers-17-01724-f002:**
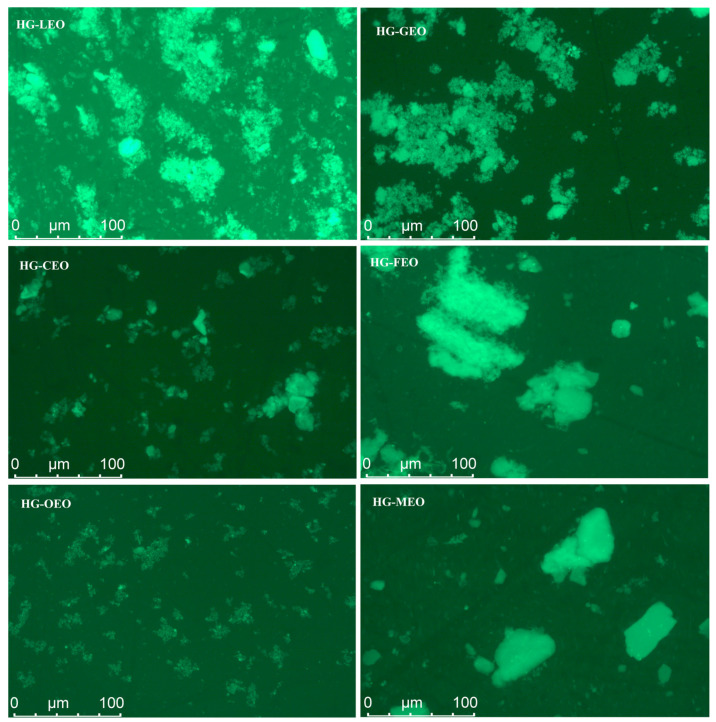
Distribution of essential oil encapsulated in hemp protein isolate (HPI)–gallic acid conjugates. HG: conjugates of hemp protein isolate and gallic acid; LEO: essential oil from lemon; GEO: essential oil from grapefruit; CEO: essential oil from camellia; FEO: essential oil from fragrans; OEO: essential oil from oregano; MEO: essential oil from mustard.

**Figure 3 polymers-17-01724-f003:**
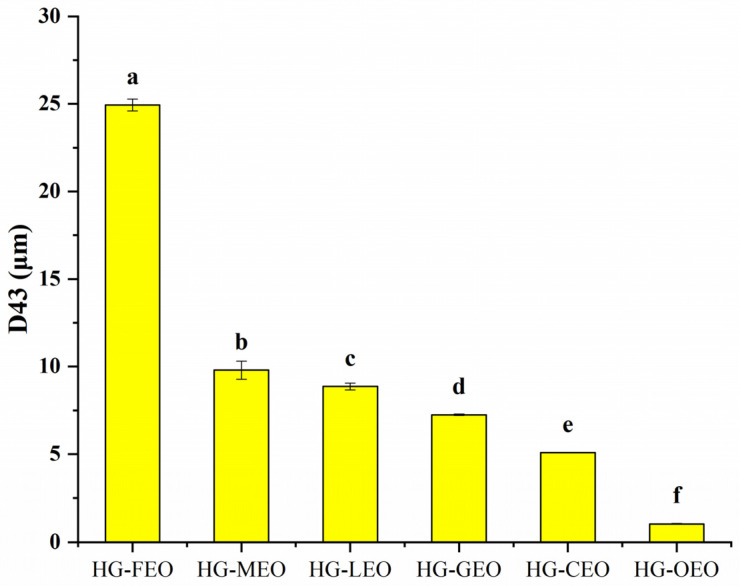
Particle size of essential oil encapsulated in hemp protein isolate (HPI)–gallic acid conjugates. HG: conjugates of hemp protein isolate and gallic acid; LEO: essential oil from lemon; GEO: essential oil from grapefruit; CEO: essential oil from camellia; FEO: essential oil from fragrans; OEO: essential oil from oregano; MEO: essential oil from mustard. Different letters mean that the values showed significant differences (*p* < 0.05).

**Figure 4 polymers-17-01724-f004:**
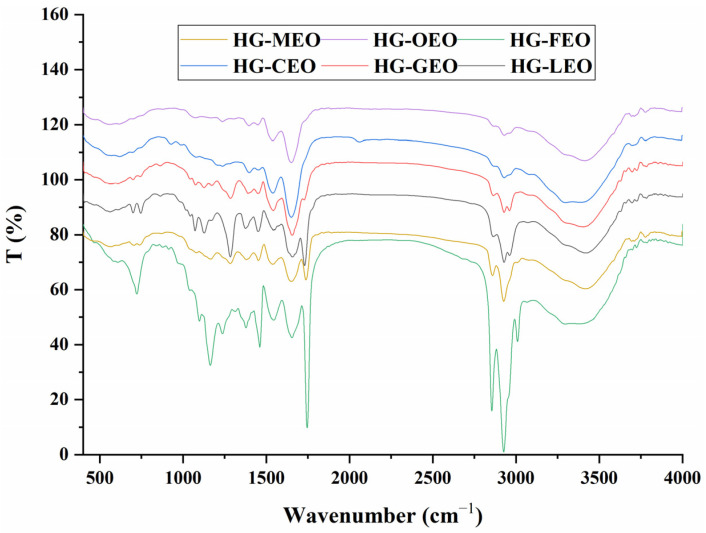
FTIR spectra of essential oil encapsulated in hemp protein isolate (HPI)–gallic acid conjugates. HG: conjugates of hemp protein isolate and gallic acid; LEO: essential oil from lemon; GEO: essential oil from grapefruit; CEO: essential oil from camellia; FEO: essential oil from fragrans; OEO: essential oil from oregano; MEO: essential oil from mustard.

**Figure 5 polymers-17-01724-f005:**
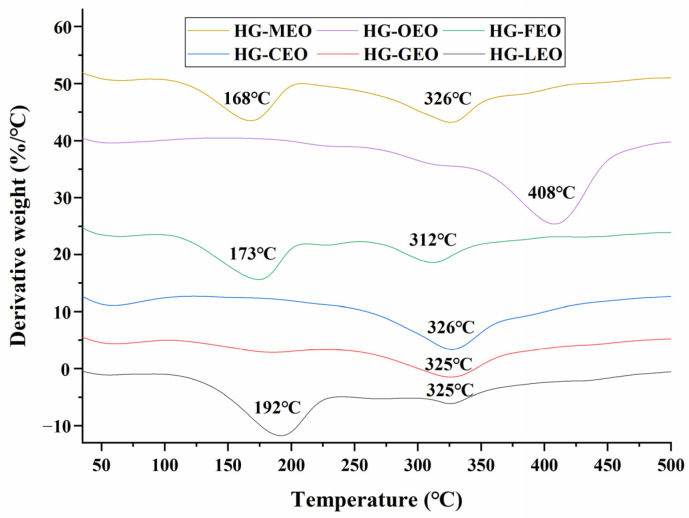
Derivative thermogravimetry curves of essential oil encapsulated in hemp protein isolate (HPI)–gallic acid conjugates. HG: conjugates of hemp protein isolate and gallic acid; LEO: essential oil from lemon; GEO: essential oil from grapefruit; CEO: essential oil from camellia; FEO: essential oil from fragrans; OEO: essential oil from oregano; MEO: essential oil from mustard.

**Figure 6 polymers-17-01724-f006:**
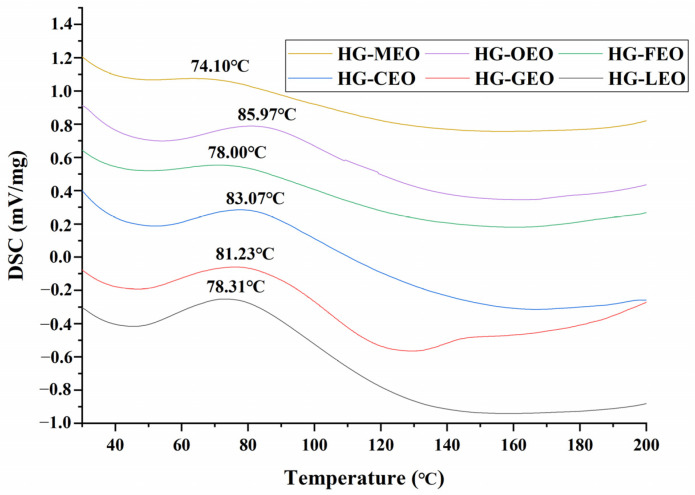
Differential scanning calorimetry curves of essential oil encapsulated in hemp protein isolate (HPI)–gallic acid conjugates. HG: conjugates of hemp protein isolate and gallic acid; LEO: essential oil from lemon; GEO: essential oil from grapefruit; CEO: essential oil from camellia; FEO: essential oil from fragrans; OEO: essential oil from oregano; MEO: essential oil from mustard.

**Figure 7 polymers-17-01724-f007:**
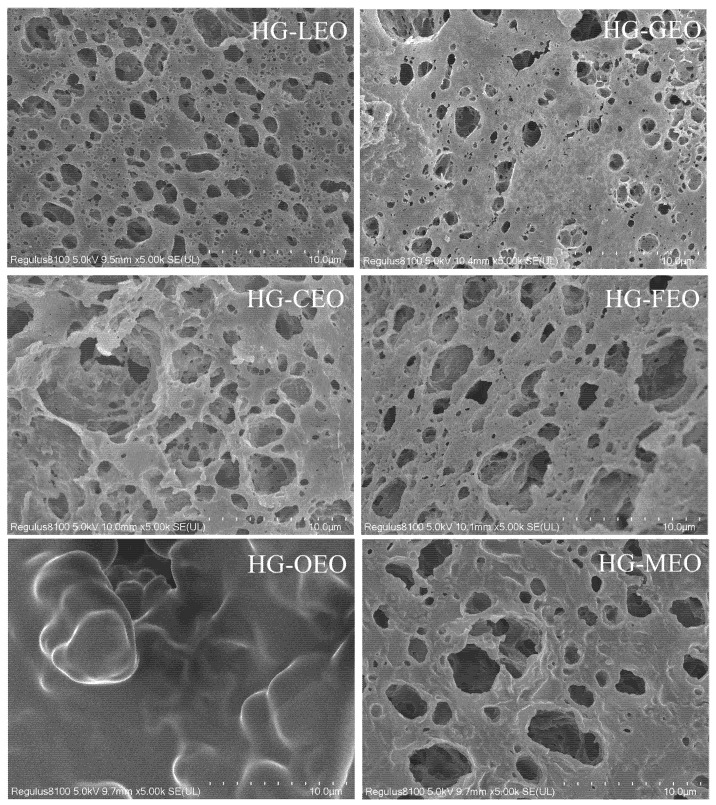
Microstructure of essential oil encapsulated in hemp protein isolate (HPI)–gallic acid conjugates. HG: conjugates of hemp protein isolate and gallic acid; LEO: essential oil from lemon; GEO: essential oil from grapefruit; CEO: essential oil from camellia; FEO: essential oil from fragrans; OEO: essential oil from oregano; MEO: essential oil from mustard.

**Figure 8 polymers-17-01724-f008:**
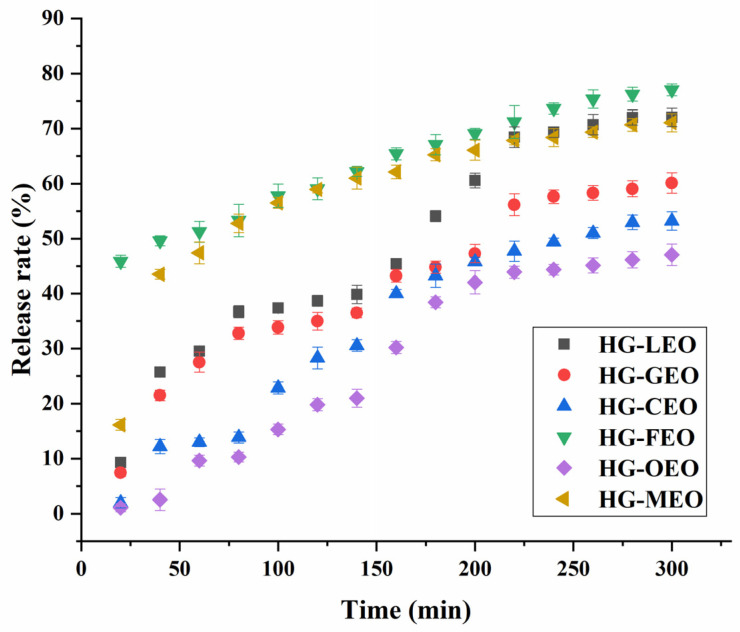
Kinetic release of essential oil from hemp protein isolate (HPI)–gallic acid conjugates. HG: conjugates of hemp protein isolate and gallic acid; LEO: essential oil from lemon; GEO: essential oil from grapefruit; CEO: essential oil from camellia; FEO: essential oil from fragrans; OEO: essential oil from oregano; MEO: essential oil from mustard.

**Figure 9 polymers-17-01724-f009:**
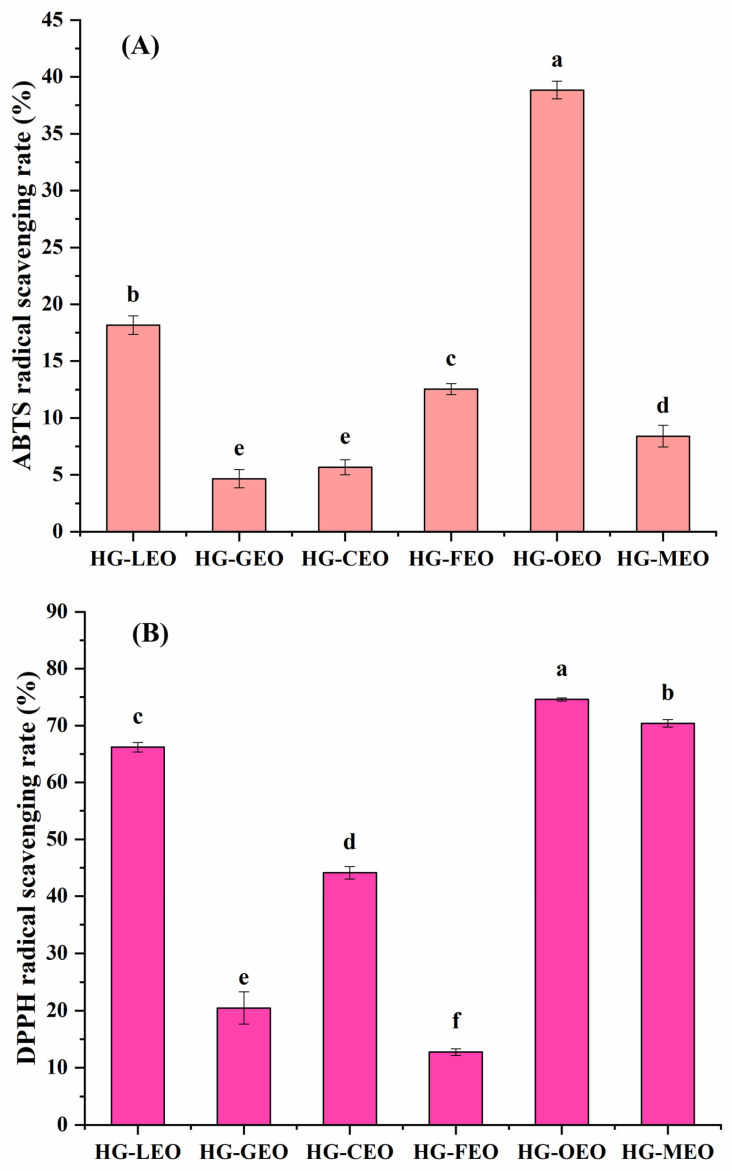
Antioxidant activity ((**A**): ABTS free radicals; (**B**): DPPH free radicals) of essential oil encapsulated in hemp protein isolate (HPI)–gallic acid conjugates. HG: conjugates of hemp protein isolate and gallic acid; LEO: essential oil from lemon; GEO: essential oil from grapefruit; CEO: essential oil from camellia; FEO: essential oil from fragrans; OEO: essential oil from oregano; MEO: essential oil from mustard. Superscripts with different letters in the same pattern are significantly different (*p* < 0.05).

**Figure 10 polymers-17-01724-f010:**
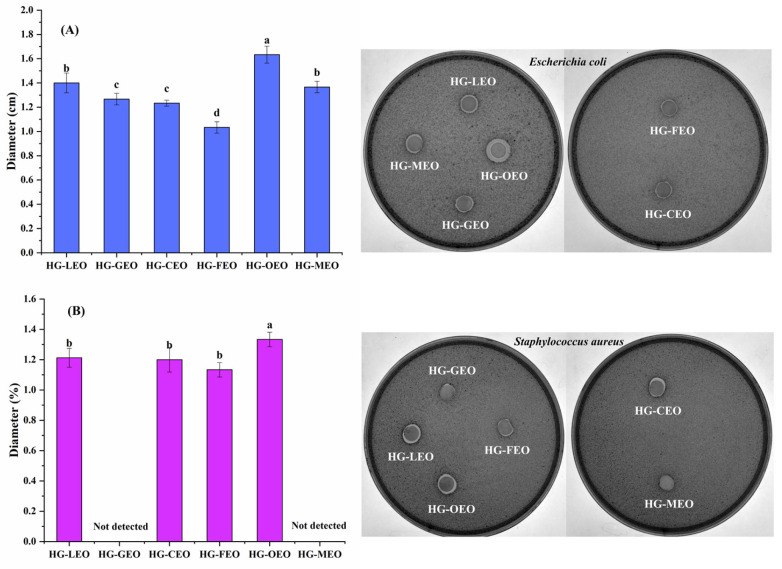
Antibacterial activity ((**A**): *Escherichia coli*; (**B**): *Staphylococcus aureus*) of essential oil-encapsulated hemp protein isolate (HPI)–gallic acid conjugates. HG: conjugates of hemp protein isolate and gallic acid; LEO: essential oil from lemon; GEO: essential oil from grapefruit; CEO: essential oil from camellia; FEO: essential oil from fragrans; OEO: essential oil from oregano; MEO: essential oil from mustard. Superscripts with different letters in the same pattern are significantly different (*p* < 0.05).

**Table 1 polymers-17-01724-t001:** Kinetic parameters of release of essential oils (EOs) from microcapsules.

Sample	Peppas Model			Weibull Model		
	R^2^	Model Constants		R^2^	Model Constants	
		Kp	n		a	b
HG-LEO	0.9590	0.0160	0.6802	0.9509	0.0083	0.8860
HG-GEO	0.9630	0.0146	0.6682	0.9674	0.0094	0.8139
HG-CEO	0.9370	0.0013	1.0970	0.9535	0.0008	1.2423
HG-FEO	0.9601	0.2249	0.2114	0.9425	0.1804	0.3548
HG-OEO	0.9077	0.0002	1.4230	0.9157	0.0001	1.5551
HG-MEO	0.8629	0.0695	0.4286	0.9352	0.0482	0.5912

HG: conjugates of hemp protein isolate and gallic acid; LEO: essential oil from lemon; GEO: essential oil from grapefruit; CEO: essential oil from camellia; FEO: essential oil from fragrans; OEO: essential oil from oregano; MEO: essential oil from mustard.

## Data Availability

All data generated or analyzed during this study are included in this manuscript.

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
