# Peer review of "Encapsulation of Essential Oils Using Hemp Protein Isolate–Gallic Acid Conjugates: Characterization and Functional Evaluation"

_polymers, 2025, doi:10.3390/polym17131724_

Round 1
Reviewer 1 Report
Comments and Suggestions for Authors
Questions.
- What is the ratio between the mass of the carrier and the essential oil it contains? In other words, how much oil is contained in 100 grams of the carrier?
- 2. Is the shelf life of such products determined? If they are used for preservation, it is important to understand how long the essential oil can remain inside the capsules. If so, how is the shelf life assessed?
Author Response
Comment 1: What is the ratio between the mass of the carrier and the essential oil it contains? In other words, how much oil is contained in 100 grams of the carrier?
Response 1: We appreciate your suggestion. As requested, we have supplemented the corresponding data in Line 235-237.
Comment 2: Is the shelf life of such products determined? If they are used for preservation, it is important to understand how long the essential oil can remain inside the capsules. If so, how is the shelf life assessed?
Response 2: We sincerely appreciate your valuable suggestion regarding the shelf-life evaluation of microcapsules. While we acknowledge the importance of shelf-life stability for practical applications, the primary focus of this study was to systematically investigate the influence of different essential oils (EOs) on the physicochemical properties of microcapsules. Our current work did not include shelf-life testing under varying storage conditions. However, as highlighted in the Conclusions section (Lines 516), we have explicitly proposed future research directions to address this critical aspect. Specifically, we plan to examine the effects of temperature, storage duration, and relative humidity on EO stability within microcapsules, which will enable predictive modeling of shelf-life behavior.
Reviewer 2 Report
Comments and Suggestions for Authors
Abstract
Clearly state the specific limitation in current encapsulation methods that this study addresses to justify the need for this work.
Define what is novel about using HPI-GA conjugates—whether it's the first time this combination is used, or if the method offers distinct advantages.
Clearly identify the target industrial applications (e.g., food preservation, pharmaceuticals) to enhance the impact and applicability of the findings.
Introduction
The knowledge gap about HPI-GA conjugates is buried late in the introduction. Move this gap closer to the beginning to establish a strong rationale for the study upfront.
Improve transitions between paragraphs—e.g., from general EO benefits to encapsulation strategies, then to material types, then to hemp protein. Currently, the flow feels slightly disjointed and overly segmented.
The section is densely packed with advanced terms (e.g., "multiprotein assemblies with tunable crosslinking density") that could be briefly defined or simplified for readability without sacrificing technical accuracy.
While limitations are stated, clarify why hemp protein is still chosen—e.g., sustainability, underexplored potential, or byproduct valorization.
The ultrasonication-induced conjugation approach is described, but its novelty relative to existing methods is not clearly highlighted. Emphasize what differentiates your method.
The importance of this research in industrial applications is mentioned but should be integrated sooner to guide the reader’s understanding of its impact.
Materials and Methods
I recommend the authors provide a brief explanation for the selection of the specific essential oils used in this study. Including this information in the Introduction or Materials and Methods section would help readers understand the rationale behind their choice, especially in relation to their known antimicrobial properties, compatibility with HPI-gallic acid conjugates, or relevance to food or pharmaceutical applications.
Provide specific catalog numbers or purity grades for critical reagents (e.g., gallic acid, essential oils, solvents) to enhance reproducibility.
Clearly define the basis for choosing specific concentrations and parameters (e.g., why 15 mg gallic acid per 1 g HPI?)
Clarify steps involving "previously established protocols" – ensure essential steps are described in full or key modifications are clearly stated.
Mention number of replicates for each experimental method (not just “triplicate” in one section).
I suggest the authors briefly justify the choice of Escherichia coli and Staphylococcus aureus in the antibacterial assay. Although these are widely used model organisms representing Gram-negative and Gram-positive bacteria, highlighting their relevance to the intended application (e.g., food safety, biomedical relevance) would strengthen the context of the study.
Results and Discussion
Some claims (e.g., the role of specific chemical components like carvacrol) are plausible but should be better substantiated with quantitative data or additional references.
The discussion could benefit from more critical comparisons between the current study’s findings and previous literature, highlighting similarities and explaining any discrepancies.
More explicit referencing of specific trends or data points in figures (e.g., Figure 1, 3, 4) would improve coherence and link the narrative to visual results more effectively.
Some essential oils (e.g., oregano) are discussed in greater detail than others. Balancing the depth of discussion would make the section more comprehensive.
For example, the unexpected FTIR signals in oregano and fragrans microcapsules are briefly mentioned but not adequately analyzed or explored for potential experimental limitations.
The current data presented in Figure 8 suggest that the essential oil release profiles, particularly for the HG-LEO sample, have not yet reached a stable plateau. The continued upward trend indicates that release is still occurring, and the endpoint of the release process has not been clearly captured. I recommended to extend the release study to include additional time points until a clear stabilization is observed. This will help accurately define the release kinetics and confirm the maximum release level. Including several more data points showing that no further release occurs would strengthen the interpretation and reliability of the release profile.
Figure 10 presents antibacterial activity data for the encapsulated essential oil formulations against E. coli and S. aureus. However, the antimicrobial performance of the unencapsulated essential oils at the same concentrations used in the encapsulated versions is not provided. Including these unencapsulated controls would offer a direct comparison to evaluate the effectiveness of encapsulation on antimicrobial activity. It is strongly recommended to present the inhibition zone diameters (in mm) for the unencapsulated essential oils under the same experimental conditions. Additionally, the inclusion of representative images of the Petri dishes showing the inhibition zones would enhance the visual clarity and scientific rigor of the data. These images could be placed in the main manuscript or in the supplementary materials.
Conclusions
The section is overly focused on summarizing detailed findings already discussed. Conclusions should synthesize insights rather than re-list specific results (e.g., microstructure, encapsulation efficiency, etc.).
Emphasize what is novel about using HPI–gallic acid conjugates as encapsulation materials compared to existing systems. This is implied but not clearly stated.
Restate how the findings address the original research aim or hypothesis presented in the introduction. Currently, there's a disconnect.
Briefly acknowledging study limitations (e.g., in vitro nature, no comparison with unencapsulated oils) would add depth and critical perspective.
The suggestion for future work is vague. Specify what parameters or environments should be tested in food systems or what challenges exist in scaling up.
Author Response
Comment 1: Abstract: Clearly state the specific limitation in current encapsulation methods that this study addresses to justify the need for this work.
Response 1: Thank you for this constructive suggestion. We have revised the Abstract to explicitly address the limitations of the microencapsulation approach and to more clearly articulate the study's objectives (Lines 12-13).
Comment 2: Abstract: Define what is novel about using HPI-GA conjugates—whether it's the first time this combination is used, or if the method offers distinct advantages.
Response 2: Thank you for this constructive feedback. We have expanded the Abstract to include specific characteristics of the hemp seed protein-gallic acid conjugates, highlighting its advantages for encapsulation purposes (Lines 14-15).
Comment 3: Abstract: Clearly identify the target industrial applications (e.g., food preservation, pharmaceuticals) to enhance the impact and applicability of the findings.
Response 3: We sincerely appreciate your constructive suggestion. In response to your comment, we have supplemented the concluding part of the Abstract with a clear statement regarding the potential application fields of this research (Lines 27-30).
Comment 4: Introduction: The knowledge gap about HPI-GA conjugates is buried late in the introduction. Move this gap closer to the beginning to establish a strong rationale for the study upfront.
Response 4: Thank you for your constructive feedback. We have carefully restructured the Introduction section, relocating the discussion of the HPI-GA conjugates to a more prominent position at the start of the fourth paragraph. Furthermore, to emphasize the importance of this conjugate system, we have added a clear description of its characteristics at the beginning of the Abstract.
Comment 5: Introduction: Improve transitions between paragraphs—e.g., from general EO benefits to encapsulation strategies, then to material types, then to hemp protein. Currently, the flow feels slightly disjointed and overly segmented.
Response 5: Thank you for your valuable feedback. We have carefully rewritten the Introduction section to enhance its logical structure and improve the connections between paragraphs.
Comment 6: Introduction: The section is densely packed with advanced terms (e.g., "multiprotein assemblies with tunable crosslinking density") that could be briefly defined or simplified for readability without sacrificing technical accuracy.
Response 6: Thank you for your valuable feedback. The “multiprotein assemblies with tunable crosslinking density” have been changed into “assemblies of multiple proteins with adjustable molecular linkages (crosslinking density)” (Line 53-54).
Comment 7: Introduction: While limitations are stated, clarify why hemp protein is still chosen—e.g., sustainability, underexplored potential, or byproduct valorization.
Response 7: Thank you for your valuable feedback. We have explicitly stated: “presents an attractive, underexploited resource offering significant potential for sustainability and waste valorization”. (Line 60)
Comment 8: Introduction: The ultrasonication-induced conjugation approach is described, but its novelty relative to existing methods is not clearly highlighted. Emphasize what differentiates your method.
Response 8: Thank you for your valuable feedback. We have added explanation of mechanism and advantages: “This approach leverages intense sound waves to generate free radicals, driving rapid covalent bond formation under mild conditions, offering distinct advantages over conventional chemical or enzymatic conjugation routes in terms of speed, efficiency, and potential for scale-up.” (Line 68-71)
Comment 9: Introduction: The importance of this research in industrial applications is mentioned but should be integrated sooner to guide the reader’s understanding of its impact.
Response 9: Thank you for your valuable feedback. We have added “valued across food, pharmaceutical, and agricultural industries” (Line 36) and “widespread industrial utilization” when stating the limitation in the first paragraph (Line 39).
Comment 10: Materials and Methods: I recommend the authors provide a brief explanation for the selection of the specific essential oils used in this study. Including this information in the Introduction or Materials and Methods section would help readers understand the rationale behind their choice, especially in relation to their known antimicrobial properties, compatibility with HPI-gallic acid conjugates, or relevance to food or pharmaceutical applications.
Response 10: Thank you for your feedback. We have carefully revised the manuscript by adding a detailed explanation regarding the selection criteria for these essential oils in the concluding paragraph of the Introduction (Line 78-81).
Comment 11: Materials and Methods: Provide specific catalog numbers or purity grades for critical reagents (e.g., gallic acid, essential oils, solvents) to enhance reproducibility.
Response 11: We sincerely appreciate your suggestions. In response to these comments, we have: Added the product identification numbers provided by the supplier for all six essential oils; Included references to published studies that utilized essential oils from the same manufacturer to facilitate experimental reproducibility; Supplemented purity information for all other chemicals as requested. (Line 91-96)
Comment 12: Materials and Methods: Clearly define the basis for choosing specific concentrations and parameters (e.g., why 15 mg gallic acid per 1 g HPI?)
Response 12: We appreciate your comment. Regarding the experimental procedures for protein extraction, conjugates production, and microencapsulation, all protocols were adapted from well-established methods in the literature. We have carefully cited the relevant references in the Methods section.
Comment 13: Materials and Methods: Clarify steps involving "previously established protocols" – ensure essential steps are described in full or key modifications are clearly stated.
Response 13: We sincerely appreciate your comment. After thoroughly re-examining the Methods section, we confirm that all critical steps and parameters from the referenced protocols have been explicitly described to ensure methodological reproducibility.
Comment 14: Materials and Methods: Mention number of replicates for each experimental method (not just “triplicate” in one section).
Response 14: We gratefully acknowledge this suggestion. The revised manuscript now clearly states that all measurements were performed in triplicate, with details provided for each analytical method.
Comment 15: Materials and Methods: I suggest the authors briefly justify the choice of Escherichia coli and Staphylococcus aureus in the antibacterial assay. Although these are widely used model organisms representing Gram-negative and Gram-positive bacteria, highlighting their relevance to the intended application (e.g., food safety, biomedical relevance) would strengthen the context of the study.
Response 15: We sincerely appreciate your valuable suggestion. In response, we have now added justification for selecting these two bacterial strains in the revised manuscript, highlighting their dual significance as both Gram-type model organisms and representative foodborne pathogens (Line 209-213).
Comment 16: Results and Discussion: Some claims (e.g., the role of specific chemical components like carvacrol) are plausible but should be better substantiated with quantitative data or additional references.
Response 16: We sincerely appreciate your valuable suggestion. In response, we have now added three key references that conclusively demonstrate carvacrol as the primary constituent of oregano essential oil (Line 250). We have also incorporated findings reporting successful preparation of amphiphilic antimicrobial agents using carvacrol as component (Line 252).
Comment 17: Results and Discussion: The discussion could benefit from more critical comparisons between the current study’s findings and previous literature, highlighting similarities and explaining any discrepancies.
Response 17: We gratefully acknowledge these suggestions. The revised manuscript now contains: 6 references on encapsulation parameters (Line 253); 3 studies supporting particle size analysis (Line 283-285); 10 new citations for bioactivity validation (Line 439-441 and 465-468).
Comment 18: Results and Discussion: More explicit referencing of specific trends or data points in figures (e.g., Figure 1, 3, 4) would improve coherence and link the narrative to visual results more effectively.
Response 18: We sincerely appreciate the reviewer's constructive suggestions. In response, we have carefully revised Figures 1, 3, and 4 to present the data more clearly. The modified figures now systematically organize the data according to numerical values (Figure 1 and 3) and characteristic absorption peak intensities (Figure 4).
Comment 19: Results and Discussion: Some essential oils (e.g., oregano) are discussed in greater detail than others. Balancing the depth of discussion would make the section more comprehensive. For example, the unexpected FTIR signals in oregano and fragrans microcapsules are briefly mentioned but not adequately analyzed or explored for potential experimental limitations.
Response 19: We sincerely appreciate your comments. Our focus on oregano essential oil microcapsules in the discussion section stems from their superior physicochemical properties, which make them particularly promising for further investigation. Regarding the unidentified spectral peaks in FTIR analysis, we acknowledge that their origins remain unclear despite our thorough examination of possible functional groups. This limitation will be addressed in our future work.
Comment 20: Results and Discussion: The current data presented in Figure 8 suggest that the essential oil release profiles, particularly for the HG-LEO sample, have not yet reached a stable plateau. The continued upward trend indicates that release is still occurring, and the endpoint of the release process has not been clearly captured. I recommended to extend the release study to include additional time points until a clear stabilization is observed. This will help accurately define the release kinetics and confirm the maximum release level. Including several more data points showing that no further release occurs would strengthen the interpretation and reliability of the release profile.
Response 20: We sincerely appreciate your valuable suggestions. In response to your comments, we have: Conducted additional release experiments with extended duration (from 240 min to 280 min) to ensure each sample reached near-equilibrium; Updated all kinetic fitting parameters accordingly.
Comment 21: Results and Discussion: Figure 10 presents antibacterial activity data for the encapsulated essential oil formulations against E. coli and S. aureus. However, the antimicrobial performance of the unencapsulated essential oils at the same concentrations used in the encapsulated versions is not provided. Including these unencapsulated controls would offer a direct comparison to evaluate the effectiveness of encapsulation on antimicrobial activity. It is strongly recommended to present the inhibition zone diameters (in mm) for the unencapsulated essential oils under the same experimental conditions. Additionally, the inclusion of representative images of the Petri dishes showing the inhibition zones would enhance the visual clarity and scientific rigor of the data. These images could be placed in the main manuscript or in the supplementary materials.
Response 21: We sincerely appreciate your valuable suggestion. In our preliminary experiments, we systematically compared the antimicrobial efficacy of free essential oils. As shown in Fig.R1 and R2, none of the six free essential oils demonstrated measurable inhibition zones against E. coli, while all their microencapsulated counterparts exhibited distinct inhibition zones (Fig. 10). Similarly, against S. aureus, four microencapsulated formulations showed clear inhibition zones (Fig. 10), whereas their non-encapsulated forms displayed no detectable activity (Fig.R3). This dramatic enhancement in antimicrobial performance following microencapsulation likely stems from two key factors: (1) the inherent high volatility of free essential oils, leading to rapid active component loss, and (2) poor compatibility with aqueous culture media resulting in inadequate dispersion. Furthermore, we have carefully addressed your suggestion by supplementing Figure 10 with high-resolution photographic documentation of the inhibition zones for microencapsulated essential oils against E. coli and S. aureus.
Fig.R1 The antibacterial activity of free essential oils against Escherichia coli. FEO: essential oil from fragrans; OEO: essential oil from oregano; MEO: essential oil from mustard.
Fig.R2 The antibacterial activity of free essential oils against Escherichia coli. LEO: essential oil from lemon; GEO: essential oil from grapefruit; CEO: essential oil from camellia.
Fig.R3 The antibacterial activity of free essential oils against Staphylococcus aureus. LEO: essential oil from lemon; CEO: essential oil from camellia; FEO: essential oil from fragrans; OEO: essential oil from oregano.
Comment 22: Conclusions: The section is overly focused on summarizing detailed findings already discussed. Conclusions should synthesize insights rather than re-list specific results (e.g., microstructure, encapsulation efficiency, etc.). Emphasize what is novel about using HPI–gallic acid conjugates as encapsulation materials compared to existing systems. This is implied but not clearly stated. Restate how the findings address the original research aim or hypothesis presented in the introduction. Currently, there's a disconnect. Briefly acknowledging study limitations (e.g., in vitro nature, no comparison with unencapsulated oils) would add depth and critical perspective. The suggestion for future work is vague. Specify what parameters or environments should be tested in food systems or what challenges exist in scaling up.
Response 22: We sincerely appreciate your valuable suggestions for improving our manuscript. In response to your recommendations, we have thoroughly revised the Conclusions section with the following key modifications: (1) We have removed the detailed reiteration of specific results to maintain focus on broader implications; (2) Added new discussion highlighting the innovative aspects of using HPI-GA covalent crosslinking products as microcapsule wall materials; (3) Included explicit statements verifying how our experimental results support the original hypothesis regarding; (4) Added a dedicated paragraph outlining study limitations; (5) Proposed concrete directions for future research to facilitate industrial translation, particularly focusing on scaling-up challenges. We hope these comprehensive revisions meet your expectations and significantly strengthen the manuscript's impact.

Reviewer 3 Report
Comments and Suggestions for Authors
In this manuscript, Xinyu Zhang et al. developed hemp protein isolate–gallic acid (HPI–GA) conjugates as a dual-functional emulsifier and wall material for encapsulating various essential oils. They evaluated their physicochemical properties, release kinetics, antioxidant capacity, and antimicrobial efficacy against foodborne pathogens. To make the review more integral, here are some issues that should be addressed.
- Did the authors compare the encapsulation efficiency and stability of HPI–GA conjugates with other commonly used materials like chitosan or whey protein?
- Did the authors perform statistical analysis to determine which model fits the data better, the Peppas or the Weibull model?
- What are the specific molecular mechanisms by which GA conjugation enhances emulsifying and encapsulating efficiency, was any structural or conformational data such as CD spectroscopy, zeta potential collected?
Author Response
Comment 1: Did the authors compare the encapsulation efficiency and stability of HPI–GA conjugates with other commonly used materials like chitosan or whey protein?
Response 1: We sincerely appreciate your valuable suggestions regarding the comparative analysis of microencapsulation performance. In response to your comments, we have compared the encapsulation efficiency of our HPI-GA microcapsules with those prepared using natural polymers, including chitosan and whey protein as wall materials. The results demonstrated comparable encapsulation efficiencies (Line 235). Regarding stability evaluation, we acknowledge that this current study did not include systematic stability comparisons between these microcapsule systems, as our primary focus was on establishing the fundamental encapsulation parameters. However, we fully recognize the importance of stability assessment, and this will be a key focus of our future research. We plan to conduct comprehensive stability testing under various environmental conditions (temperature, humidity, and storage time) to thoroughly evaluate and compare the performance of these different microencapsulation systems. These additional stability studies will provide more complete data for practical applications and industrial translation. We have added this important limitation and future direction to the revised Conclusion section.
Comment 2: Did the authors perform statistical analysis to determine which model fits the data better, the Peppas or the Weibull model?
Response 2: We sincerely appreciate your insightful question regarding our approach to release kinetics modeling. In our study, we chose to use the mean values of release data for kinetic fitting based on careful methodological considerations. This decision was made because preliminary analyses revealed significant variability among individual measurements, where random selection of single data points could substantially influence the fitting outcomes. By employing averaged data, we aimed to minimize this selection bias while preserving the overall release profile characteristics. For model selection, we primarily relied on correlation coefficients.
Comment 3: What are the specific molecular mechanisms by which GA conjugation enhances emulsifying and encapsulating efficiency, was any structural or conformational data such as CD spectroscopy, zeta potential collected?
Response 3: We sincerely appreciate your interest in the mechanism behind GA-induced improvement of HPI's emulsifying properties. As you correctly noted, this was indeed the primary focus of our previous study (https: //doi.org/ 10.1016/ j.fufo. 2022. 100210). In that work, we systematically demonstrated that covalent conjugation with gallic acid (particularly through ultrasound-assisted treatment) significantly enhanced the emulsifying properties of hemp protein isolate (HPI). These improvements were mechanistically linked to several structural modifications: (1) partial protein unfolding, (2) reduced particle size, (3) increased surface hydrophobicity, (4) lower interfacial tension, and (5) enhanced charge intensity. We have referenced and briefly summarized these key findings in the Introduction section (Line 72) of the current manuscript to provide proper context, while focusing our present work on the novel microencapsulation applications of this conjugate.
Round 2
Reviewer 2 Report
Comments and Suggestions for Authors
Thank you for submitting the revised version of the manuscript. The changes and additions you have made have significantly improved the clarity, quality, and overall presentation of the study. I appreciate your thoughtful responses to the comments and your efforts to strengthen the manuscript.
Thank you for addressing my previous suggestion and extending the release study duration. The addition of these extra data points has certainly improved the reliability of the release profiles. However, I noticed that some treatments still exhibit a continued release trend beyond 280 minutes. For example, the blue data points in the graph appear to indicate that release is still ongoing and has not yet stabilized. To fully capture the stabilization phase and accurately define the release kinetics, it would be beneficial to extend the release duration further and include additional time points. This would enhance the robustness of the interpretation and strengthen the conclusions regarding release behavior. If you have already collected additional data beyond 280 minutes, I encourage you to include them in the graph to provide a more comprehensive view of the release profiles. Also, please ensure that the kinetic fitting parameters are updated accordingly if any new data points are added.
In the current release profiles, the maximum release observed appears to range between approximately 45% and 70%, depending on the treatment. Could the authors please clarify what is happening to the remaining portion of the encapsulated compound? Is the release system designed to achieve partial release only, or are there limitations (e.g., matrix retention, degradation, or analytical constraints) preventing full release? A brief discussion on whether 100% release is expected—or theoretically possible—for these systems would help readers better understand the efficiency and practical application of the encapsulation and release approach used in this study.
I recommend that you provide a brief comparison of your release profiles—particularly the duration and extent of essential oil release—with those reported in similar studies. This will help readers better understand how your system performs relative to existing delivery systems. For example, is the release duration in your study relatively short, extended, or comparable to others using similar biopolymer-based matrices?
Author Response
Comment 1: Thank you for addressing my previous suggestion and extending the release study duration. The addition of these extra data points has certainly improved the reliability of the release profiles. However, I noticed that some treatments still exhibit a continued release trend beyond 280 minutes. For example, the blue data points in the graph appear to indicate that release is still ongoing and has not yet stabilized. To fully capture the stabilization phase and accurately define the release kinetics, it would be beneficial to extend the release duration further and include additional time points. This would enhance the robustness of the interpretation and strengthen the conclusions regarding release behavior. If you have already collected additional data beyond 280 minutes, I encourage you to include them in the graph to provide a more comprehensive view of the release profiles. Also, please ensure that the kinetic fitting parameters are updated accordingly if any new data points are added.
Response 1: We sincerely appreciate your valuable suggestion regarding the selection of release time points. In our study, we chose 280 minutes as the endpoint for release kinetics analysis based on rigorous statistical evaluation. Comparative analysis of release data at 260 minutes and 280 minutes showed no significant difference, indicating that essential oil release had essentially reached equilibrium by 260 minutes. To further validate this observation, we conducted extending release measurements up to 300 minutes, which similarly demonstrated no statistically significant differences compared to 280-minute data points. Following your suggestion, we have now incorporated the 300-minute release data in the revised Figure 8 to provide a more comprehensive representation of the release profile. Additionally, we have updated the release model parameters in Table 1 to reflect this extended dataset.
Comment 2: In the current release profiles, the maximum release observed appears to range between approximately 45% and 70%, depending on the treatment. Could the authors please clarify what is happening to the remaining portion of the encapsulated compound? Is the release system designed to achieve partial release only, or are there limitations (e.g., matrix retention, degradation, or analytical constraints) preventing full release? A brief discussion on whether 100% release is expected—or theoretically possible—for these systems would help readers better understand the efficiency and practical application of the encapsulation and release approach used in this study.
Response 2: We sincerely appreciate your valuable suggestion. In Section 3.7 of our revised manuscript, we have specifically emphasized that all observed essential oil release rates in our system remained below 80 %. We have conducted a discussion of this phenomenon. We have explicitly acknowledged these limitations in the revised section 3.7, where we also outline promising future research directions to address these challenges. (Line 410-417)
Comment 3: I recommend that you provide a brief comparison of your release profiles—particularly the duration and extent of essential oil release—with those reported in similar studies. This will help readers better understand how your system performs relative to existing delivery systems. For example, is the release duration in your study relatively short, extended, or comparable to others using similar biopolymer-based matrices?
Response 3: We sincerely appreciate your valuable suggestion. In Section 3.7 of our revised manuscript, we have conducted a comprehensive comparison of essential oil release characteristics among different protein-based microencapsulation systems, including hemp protein isolate, plum seed protein isolate, and soy protein isolate microcapsules (Line 399-406). Furthermore, we extended this comparison to protein film systems (Line 407-409).